# Therapeutic Strategies Targeting Respiratory Recovery after Spinal Cord Injury: From Preclinical Development to Clinical Translation

**DOI:** 10.3390/cells12111519

**Published:** 2023-05-31

**Authors:** Pauline Michel-Flutot, Michael A. Lane, Angelo C. Lepore, Stéphane Vinit

**Affiliations:** 1END-ICAP, UVSQ, Inserm, Université Paris-Saclay, 78000 Versailles, France; pauline.michel78280@yahoo.fr; 2Department of Neuroscience, Jefferson Synaptic Biology Center, Vickie and Jack Farber Institute for Neuroscience, Sidney Kimmel Medical College, Thomas Jefferson University, Philadelphia, PA 19107, USA; angelo.lepore@jefferson.edu; 3Marion Murray Spinal Cord Research Center, Department of Neurobiology and Anatomy, Drexel University College of Medicine, Philadelphia, PA 19129, USA; mlane.neuro@gmail.com

**Keywords:** spinal cord injury, therapeutics, preclinical models, human, respiratory

## Abstract

High spinal cord injuries (SCIs) lead to permanent functional deficits, including respiratory dysfunction. Patients living with such conditions often rely on ventilatory assistance to survive, and even those that can be weaned continue to suffer life-threatening impairments. There is currently no treatment for SCI that is capable of providing complete recovery of diaphragm activity and respiratory function. The diaphragm is the main inspiratory muscle, and its activity is controlled by phrenic motoneurons (phMNs) located in the cervical (C3–C5) spinal cord. Preserving and/or restoring phMN activity following a high SCI is essential for achieving voluntary control of breathing. In this review, we will highlight (1) the current knowledge of inflammatory and spontaneous pro-regenerative processes occurring after SCI, (2) key therapeutics developed to date, and (3) how these can be harnessed to drive respiratory recovery following SCIs. These therapeutic approaches are typically first developed and tested in relevant preclinical models, with some of them having been translated into clinical studies. A better understanding of inflammatory and pro-regenerative processes, as well as how they can be therapeutically manipulated, will be the key to achieving optimal functional recovery following SCIs.

## 1. Introduction

Spinal cord injuries (SCIs) have an occurrence of approximately 700,000 new cases worldwide each year [1], and they lead to sensory, motor, and autonomic deficiencies [2,3]. In most cases, SCIs occur in high spinal cord segments, which compromises respiratory function. Patients with mid-to-high cervical injuries often rely on ventilatory assistance to survive, but this significantly increases the risk of respiratory infections and pneumonia. Even those that recover voluntary control of breathing are at risk of respiratory failure. Accordingly, impaired breathing after a cervical SCI is a leading cause of death for these individuals [4,5]. To date, no therapies can completely restore respiratory function post-SCI [6,7,8,9], but some limited spontaneous recovery does occur, which can be attributed to neuroplasticity occurring through the remaining spinal fibers [10]. There is an urgent need for new therapeutics to improve respiratory recovery; a number of laboratories worldwide are working on this critically important problem. In this review, we will provide an overview of the key therapeutics that have been developed and tested in relevant preclinical SCI models, as well as the treatments that have been translated into human studies. All of these therapeutics are based on a comprehensive understanding of the deleterious and beneficial processes occurring post-SCI. Indeed, understanding and harnessing these diverse inflammatory and neuroplastic mechanisms will likely be a key to effective spinal cord repair.

### 1.1. Inflammation

The initial mechanical trauma of an SCI results in extensive tissue destruction at the level of injury, including vascular compromise, cell loss, and direct disruption of neural pathways [11]. These initial events are followed by a cascade of inflammatory and other pathological processes, which are collectively referred to as the secondary injury [11]. This includes vascular trauma and hemorrhage, which lead to progressive blood and fluid accumulation (edema) [12]. Ischemic damage of intact tissues from blood vessel disruption and any potential tissue compression (e.g., from surrounding bone) contributes to additional cell death and tissue loss. This can be exacerbated if sympathetic descending pathways are impacted and vascular function is impaired [13].

There is also a rapid increase in the concentration of damage-associated molecular patterns (DAMPs), which are endogenous molecules that are released by damaged cells [14]. They activate inflammatory pathways by stimulating pro-inflammatory cytokine production by glial cells [14,15]. For example, adenosine triphosphate (ATP) is a DAMP that acts as a chemoattractant for microglia through purinergic receptors, promotes migration to the injured area, and limits the expansion of the injury to surrounding intact tissues [16,17,18]. Astrocytes and oligodendrocytes also express purinergic receptors, and their activity seems to be impacted by high levels of ATP through changes in intracellular Ca^2+^ concentrations [19,20]. An increase in the extracellular concentration of excitatory amino acid neurotransmitters such as glutamate was also observed. The subsequently repeated overactivation of the receptors for these neurotransmitters induces excitotoxicity, leading to neuronal death and activation of glial cells [21,22]. This leads to increased production of free radicals and related oxidative stress [23], which contributes to the increase in ATP concentration that was described previously [21]. Axotomy of neurons after the initial injury leads to the formation of dysfunctional dystrophic end bulbs. Axonal degeneration is accompanied by secondary demyelination and release of myelin debris at the lesion site, as well as in the perineuronal area—mainly due to Wallerian degeneration—and subsequent oligodendrocyte death [19,21].

All of these extracellular signals lead to the activation and recruitment of immune cells. Resident microglia are the first to be activated and start to release pro-inflammatory cytokines, such as interleukin (IL)-1-β, tumoral necrosis factor α, and IL6 [24,25]. Neutrophils are then recruited to the lesion site and peak around 24 h post-injury [26]. Lymphocytes [27,28], as well as monocytes [27], start to invade the injured spinal cord during the following days, and then monocytes polarize into pro- and anti-inflammatory macrophages [29,30]. Astrocytes exert diverse functions in the healthy CNS, such as the regulation of blood flow, ion, and neurotransmitter levels, energy provision, homeostasis of extracellular fluid, and synapse function and remodeling [31]. Under pathological conditions such as SCIs, they undergo a process called “reactive astrogliosis” and change their phenotype [32]. The recruitment and activation of immune and glial cells—especially of reactive astrocytes and invading fibroblasts—then leads to the formation of a fibro-glial interface, often referred to as a ‘scar’. This process, which is now broadly viewed as a wound-healing response, has been well described in several reviews [19,33,34,35,36]. This fibro-glial ‘scar’ tissue persists and contributes to chronic inflammation, as well to the release of molecules that can inhibit adaptive neuronal plasticity. These inflammatory processes and the diverse molecules released represent targets for SCI repair. For example, these processes occur around the phMN area (C3–5), which is the main topic of this review (Figure 1).

### 1.2. Plasticity-Supporting and Inhibitory Factors

#### 1.2.1. Plasticity-Supporting Molecules

Neuroinflammatory processes and the various molecules released have an impact on the composition of the perineuronal net after an SCI. This leads first to modifications of extracellular matrix properties [35,36,37]. Several extrinsic molecules—mainly growth factors—are secreted in an effort to acutely “repair” damaged tissue (within hours) following an SCI. Expression of neurotrophins [38,39] that transform growth factor beta and basic fibroblast growth factor [40], for example, has been shown to increase, possibly in an attempt to mimic an ontogenic environment and, therefore, support plasticity-like processes. The most studied neurotrophins are neurotrophin-3 (NT-3), brain-derived neurotrophic factor (BDNF), and nerve growth factor (NGF), which have the highest affinity for tyrosine kinase receptors (Trk): TrkA, TrkB, and TrkC, respectively [38,39,41]. The binding to their Trk receptors leads to the intracellular signaling that is involved in neuronal survival, axonal growth, and synaptic plasticity [42]. It has been demonstrated that NGF induces robust axonal sprouting of nociceptive neurons [43]. NT-3 is mainly involved in neuronal survival, as well as sensory and sympathetic neurite outgrowth [38,44]. BDNF has a neuroprotective role and supports axon and dendrite growth after SCIs [45]. In addition, some matrix metalloproteases, such as MMP3 and MMP7, can act to facilitate plasticity processes following SCIs through their function in cleaving pro-neurotrophic molecules in their mature form [38]. Several extracellular matrix molecules, such as collagen, fibronectin, and laminin, also support axonal regrowth [9].

Along with the vast array of extrinsic factors, intrinsic cellular factors supporting axonal growth are also expressed after SCIs. Interestingly, these intrinsic factors differ depending on the neuron subtype and the distance of axon damage from the cell body. For example, rubrospinal and corticospinal neurons upregulate regeneration-associated-gene transcription only when axotomy is near the cell body [46,47]. On the contrary, long descending propriospinal neurons preferentially upregulate these genes when the axotomy occurs far from the cell body [48]. The growth-associated protein-43 (GAP-43) is expressed in the cell body and axon membranes and is involved in neuroplastic processes [49,50]. Following SCIs, GAP-43 mRNA is upregulated in axotomized neurons [51], and its expression is also increased in deafferented phMNs in rats [52]. The transcription factors involved in neuronal plasticity and cell activity also increase in expression/activity after an injury. These include cAMP response element-binding protein (CREB) (whose phosphorylation levels correlate with increased expression of BDNF and its receptor TrkB [53]), as well as c-Jun and its phosphorylated form [54].

#### 1.2.2. Inhibitory Molecules

As described above, some damaged axons initially display a growth response following SCIs, but their regrowth is chronically aborted with the formation of scar tissue. Indeed, some molecules are expressed in the perineuronal net following SCIs and act as chemorepulsive factors for axon growth cones. These molecules, such as DAMPs, Nogo-A, semaphorin 3, Slit, and chondroitin sulfate proteoglycans (CSPGs), therefore, participate in the inhibition of synaptic plasticity after injury [15,33,34], with the latter being one of the most repulsive molecules [36,55].

In normal physiological conditions, CSPGs are involved in regulating plasticity and preventing maladaptive growth in the adult nervous system [55,56]. They are composed of a core glycoprotein along with glycosaminoglycan (GAG) sugar chains that are covalently attached [57]. Fitch and Silver showed in 1997 that reactive astrocytes present in an injured area produced CSPGs, which inhibited neurite outgrowth in vitro [58]. These CSPGs show differential expression depending on the time post-injury [59]. Their GAG chains interact with several high-affinity receptors present on the axonal growth cone membrane: protein tyrosine phosphatase σ (PTPσ), leukocyte common antigen-related phosphatase (LAR), and Nogo 1 and 3 receptors (NgR) [60,61,62]. The pathways activated by these receptors then modulate the activity of a number of signaling molecules and transcription factors, resulting in axonal growth inhibition [19,36,63].

Other types of molecules also inhibit growth and plasticity after SCIs. Some of these molecules are found in myelin and can also be expressed by neurons, including Nogo-A, myelin-associated glycoproteins (MAGs), and myelin oligodendrocyte glycoprotein [19,36]. Nogo-A binds to the NgR1 [64], Nogo-66 [64], and PirB [65] receptors, leading to actin cytoskeleton destabilization through RhoA/ROCK signaling pathway activation and the consequent arrest and collapse of axonal growth cones [64]. The molecules involved in axon guidance during development are also involved in this process. For example, chemorepulsive molecules such as Semaphorin 3A and 7 have an inhibitory effect on axonal growth [66,67,68]. The Slit/Roundabout and ephrin signaling pathways involved in axonal guidance during development also play a role in this inhibition [69,70]. Robo receptors are expressed on the membrane of axonal growth cones and use the same RhoA pathway as NgR [71], and Slit molecules are expressed by monocyte-derived macrophages and activated microglia [72]. Neuronal-intrinsic cellular factors, such as phosphatase and tensin homolog (PTEN), are also involved in this inhibition. Neuronal PTEN expression is increased in the adult CNS and after SCIs [73]. PTEN inhibition of Akt activity (which normally activates signaling axes, such as the mTOR pathway) plays a major role in limiting axonal growth capacity [74].

### 1.3. From Preclinical Models to Humans

Growth-inhibitory molecules limit repair post-SCI even in the presence of plasticity-supporting molecules (e.g., growth factors). While some treatments aim to overcome this by enhancing the expression of plasticity-supporting molecules, the extent of growth that is achievable remains limited. Instead, a key feature of these strategies is that they rely on spared tissue, cells, and axonal pathways. To test these therapeutics for respiratory recovery, two preclinical models of cervical SCIs were used: contusion injury—often at mid-cervical levels [75,76,77,78,79,80,81,82]—and high cervical hemisection—usually at the second cervical segment [83,84,85,86,87,88,89,90,91,92]. Contusion paradigms better model human pathophysiology but are often associated with greater variability in anatomical and functional outcomes. Contusive injuries were performed rostral to C3 to deafferent phMNs (located in the C3–C5 spinal cord in most mammalian species) or at the level of phMNs to directly damage them. Notably, most SCIs occur clinically at mid-cervical (C3–5) levels. Hemisection models are mainly used to deafferent phMNs and allow a high level of reproducibility to study specific plasticity processes or molecular pathways in response to injuries. The C2 hemisection model—which has been used for nearly 150 years—has the ability to reveal crossed phrenic pathways (decussating at the spinal midline) [10,93,94,95] that can be used to support diaphragm activity recovery. These two models of high SCIs are the most commonly used, and the choice of the model should be carefully considered depending on the purpose of the study. These preclinical models and others have been used to test the effects of a range of therapeutics on functional recovery, including respiratory function. Decades of preclinical work and several important discoveries continue to shape the future of promising therapeutics for SCIs and clinical translations.

## 2. Activity-Based Therapeutics

### 2.1. Intermittent Hypoxia

Activity-based therapies (e.g., rehabilitation) have been extensively used to increase the extent and rate of recovery after SCIs. These neuroplasticity-inducing treatments can promote some axonal regrowth (albeit limited), but they also largely rely on pathways spared by the injury. After cervical SCIs, these approaches have been used to amplify plasticity and motor recovery in damaged respiratory networks. Activity-based therapies are often favored as a treatment approach, as they are readily translatable and relatively non-invasive. To target respiratory activity, one strategy has been to expose the subject (pre-clinical animals or humans in some clinical studies) to respiratory challenges, such as hypoxia, which consists of a reduction in the percentage of oxygen in inspired air (normally composed of 21% oxygen) (Figure 2). The ventilatory response to hypoxia depends on the protocol’s duration and the application pattern (intermittent or continuous) and severity [96,97,98,99,100].

The use of intermittent hypoxia (IH) was pioneered by Gordon Mitchell and colleagues, who were originally at the University of Wisconsin and are now at the University of Florida. Following a chronic C2 lateral spinal cord hemisection in rats, a chronic IH protocol (5 min alternating between 11% oxygen and normoxia) was applied for 12 h per night for 7 consecutive days [101]. This protocol led to an increase in phrenic activity on the injured side compared to non-treated animals, suggesting that this protocol strengthened crossed phrenic pathways [101]. Some studies, however, demonstrated that repeated severe/chronic IH protocol application (2–8% oxygen) with a high cycle number could lead to deleterious side effects, such as hypertension [102] or hippocampus apoptosis [103]. It was then found that the use of more moderate protocols (9–16% oxygen) delivered at lower cycle numbers provided beneficial outcomes without adverse side effects [104].

Studies then focused on the application of moderate IH following high SCIs to restore respiratory function. Mitchell and his collaborators were the first to show that acute IH (AIH) enhances respiratory motor output in a rat preclinical model of SCI [105]. Subsequent studies then showed that the application of daily moderate AIH protocols led to persistent respiratory function improvement in C2 hemisected rats [106,107], which was associated with increased expression of BDNF and TrkB [107]. These beneficial effects on respiratory function were confirmed in several studies after C2 hemisection in rats [105,106,108,109,110,111,112] and mice [113], as well as in a cervical contusion model in rats [76,77,114,115]. AIH protocols also impact the expression of several molecules that trigger neuronal plasticity after SCIs, including decreased PTEN expression and increased expression in mTOR, as well as increased S6 and c-Fos expression by phMNs [116]. Recently, it was shown that IH protocols led to increased serotonergic axon density around phMNs, which could support diaphragm activity restoration after SCIs [117].

Daily AIH protocols, therefore, appear to be a safe and non-invasive way to improve respiratory function following high SCIs, and this has been extensively reviewed [8,118,119,120]. These preclinical studies demonstrated the feasibility and safety of repeated IH protocol application at low doses. IH protocols also improve respiratory plasticity and recovery, though the improvement is greater when applied acutely after injury rather than at more chronic time points [8].

Respiratory training approaches are also translatable to the clinic [104]. In individuals with chronic (>1 year) incomplete cervical SCIs, daily AIH protocol application led to increased minute ventilation during the first two days, corresponding to long-term respiratory facilitation. This effect could be maintained through the protocol’s duration, but without any changes over time [121]. In similar cases, AIH protocols seemed to have some global beneficial effects on respiratory output, although complete recovery has not yet been observed [122,123].

Ongoing pre-clinical studies have also begun to show that the observed beneficial effects can be amplified in combination with other therapeutics, such as the delivery of chondroitinase ABC (an enzyme that digests GAGs from CSPGs) [124] or pre-AIH prednisolone (anti-inflammatory corticosteroid) administration [125]. Daily AIH has also been combined with other therapeutics in clinical studies to maximize efficacy, but so far, this was only in trials aimed at improving limb motor function in people with lower SCIs [126,127,128,129]. These effects can be maximized when combined with those of other therapeutics; it is now critical to investigate their impacts on respiratory recovery after SCIs.

### 2.2. Exercise

Another form of activity-based therapy used to improve respiratory function is non-respiratory training. Locomotor training has been extensively used to improve hindlimb function in preclinical models and clinical cases following SCIs. Interestingly, these approaches can also can be used to “train” the respiratory system, as exercise induces an increased oxygen demand (Figure 2), and there is respiratory adaptation to locomotor exercise through neuronal interconnections between the respiratory and locomotor circuits [130].

Preclinical studies aimed at promoting respiratory recovery through exercise are limited. Nevertheless, it was shown that in C2 hemisected mice, chronic forced training led to increased fatigue resistance, some muscle (limb and diaphragm) plasticity, and enhanced running capacity in the trained SCI group [131].

In patients, respiratory muscle training showed beneficial outcomes after cervical SCIs [132,133], as evidenced, for example, by increased vital capacity and inspiratory reserve volume [134]. The effects of this type of training have been more extensively studied in athletes [132,135,136]. These training protocols allow for an increased maximum volume of inspired and expired air [135,136]. Vocal exercise, such as singing, is also a studied therapeutic strategy. However, the beneficial effects of such methods for SCIs remain to be determined [137]. Locomotor training has also been shown to have beneficial effects on respiratory function. For example, locomotor training with body-weight support in patients with chronic cervical SCIs led to improved respiratory function [138]. However, the beneficial effects of these exercise protocols remain limited, and more investigation is required to determine the best use of these therapeutic strategies.

## 3. Stimulation-Based Therapeutics

While activity-based therapies can non-invasively drive activity in neural networks, they tend to activate many networks and, therefore, lack precision in some regards. Neural stimulation can also be achieved by providing an exogenous electrical or chemical stimulus to directly activate neural networks (above the stimulation threshold) or modulate neural activity by providing low levels of stimulation. Broadly, these approaches can be divided into invasive or relatively non-invasive strategies.

### 3.1. Invasive Stimulation

Invasive neural stimulation strategies are exemplified by the implantation of electrodes (individually or in electrode arrays) that interface with neural tissues, nerves, or the muscles that they innervate (e.g., brain–computer interface (BCI)/brain–machine interface (BMI)). They can range from the use of implantable wires in the brain or spinal cord tissue (e.g., intraspinal microwires, Utah array) to the placement of electrodes on the surface of neural tissues (epidural stimulation). While there is an increasing number of reviews on this subject, the following section will highlight the use of epidural stimulation as a potential strategy for enhancing the recovery of breathing post-SCI.

Epidural electric stimulation (EES) directly and repeatedly stimulates spinal networks, usually at sub-threshold levels, to induce neuronal plasticity by modulating neuronal excitability (Figure 2; essentially acting as an amplifier for incoming signals and making spinal networks more likely to be active). This technique was initially used to reduce chronic pain in patients with SCIs by reducing neuronal excitability and related pain signaling [139,140]. EES has now been demonstrated to improve locomotor function in preclinical models of SCIs [141,142,143] and in humans [144,145]. In the respiratory system, EES has been used to induce neuroplasticity processes at the level of deafferented phMNs. Studies show some beneficial effects of EES application after high SCIs in preclinical models [85,146,147,148,149]. Repeated EES application (inspiration declenched; approximately 0.5 Hz; 24 h per day for 4 days) at the cervical level in awake rats following C2 hemisection led to increased BDNF and vascular endothelial growth factor (VEGF) expression in the ventral C3–C5 spinal cord where phMNs reside [149]. EES at the C4 level at 100–300 Hz led to short-term facilitation of the phrenic motor system after a C1 transection in anesthetized ventilated rats [85]. Likewise, EES potentiated ipsilateral phasic inspiratory activity at both 2 and 12 weeks after C2 hemisection when delivered at 300 Hz [147].

EES trials have also been shown to improve function in people with SCIs [150], including by improving breathing [150]. However, further investigation is required to better understand its effects and how to apply this therapeutic in the most efficient way, as complete respiratory recovery would likely not be achieved when delivering therapeutics based only on EES.

Despite the demonstration of its therapeutic efficacy, epidural stimulation is among the more invasive approaches, and ongoing work aims to assess whether less invasive approaches can yield similar levels of benefit.

### 3.2. Non-Invasive Stimulation

Several non-invasive stimulation procedures following cervical SCIs have been tested. For example, trans-spinal direct current stimulation (TDCs) has proven to induce beneficial functional improvements [151]. Cervical trans-spinal direct current stimulation could also address the loss of respiratory function in humans [152].

One of the most well-characterized non-invasive methods of neural activation is the use of magnetic stimulation (MS). MS, and more specifically repetitive MS (rMS), applies a magnetic field above target cells from outside the body (through intact bone, muscle, and skin) to neuromodulate neuronal excitability (Figure 2). Transcranial rMS was first used clinically to treat depression [153,154,155] or post-traumatic brain disorders [156,157]. In the context of SCIs, rMS has been recently used to reduce spasticity and neuropathic pain, as well as to promote functional motor recovery, in patients, mainly through cortical inhibition [158,159,160,161,162]. This approach, however, has not been used to promote respiratory recovery following cervical SCIs in patients, despite the potential efficacy considering its effects on other motor systems.

MS can be used to evaluate respiratory supraspinal plasticity in clinical studies [163,164,165,166,167,168], as well as in preclinical models [169,170,171]. In addition, a single MS burst can be a useful non-invasive tool for evaluating variations in phrenic system excitability. Concerning the effect of rMS on the respiratory system following high SCIs, it was recently shown that the application of chronic high-frequency repetitive transcranial MS in rats led to some respiratory improvement post-SCI. This strategy also elicits neuronal plasticity with a reduction in deleterious post-traumatic inflammatory processes in the cervical spinal cord [172].

Additional work is required to understand the effects of rMS after SCIs and to determine appropriate protocols for achieving optimal recovery. This new technique is promising, since it is non-invasive and broadly used in clinics to treat other central nervous system pathologies.

## 4. Therapeutics for Inducing Regeneration/Reorganization of Neural Pathways

### 4.1. Cell Transplantation

Cell therapy for SCI treatment consists of a vast range of strategies that target a number of goals, including neuroprotection, immune modulation, and the construction of new neuronal networks (Figure 2) [173,174]. Depending on the goal, different types of cells have been utilized by using a variety of delivery methods, such as intravenous injection, transplantation directly into the injured spinal cord, or injection into spinal tissues away from the injury (e.g., the level of deafferented motoneurons). Cell therapies have been tested in a host of preclinical SCI models, improving outcome measures in many cases. For example, improvements of locomotor [175,176], autonomic [177], and respiratory [7,173] function were achieved following cell transplantation into the lesioned spinal cord. A number of different donor cell types were tested after high SCIs to target respiratory recovery [7].

Because astrocytes have important and diverse beneficial roles, such as the regulation of extracellular ion and neurotransmitter homeostasis [31] and synaptogenesis [178], there is great interest in astrocyte transplantation following SCIs [179,180,181]. In contrast to the astrocytes that contribute to fibro-glial scars post-injury, donor astrocytes are pro-reparative and promote tissue repair. Astrocytes derived from sources such as human induced pluripotent stem cells injected into rats and mice at the level of SCIs induce beneficial outcomes, such as a reduced lesion area and increased diaphragm activity on the injured side [182]. Similarly, transplantation of glial-restricted precursors (a class of lineage-restricted astrocyte progenitors) into the lesion site acutely after C2 hemisection led to the partial recovery of diaphragm electromyography amplitude on the injured side, as well as the regeneration of bulbospinal respiratory axons [183].

Neuronal transplantation can also be used to repair a damaged spinal cord, for example, by generating new circuits that can relay information from injured neurons rostral to the injury to deafferented neuron populations caudal to the injury [174,184]. Depending on the donor neuron type, they can also provide neurochemical inputs into damaged spinal networks to modulate activity. Transplantation of fetal spinal cord tissue enriched in neuronal and glial precursor cells led to phrenic nerve activity recovery, which was associated with new connectivity between donor neurons and the host phrenic circuitry [185]. This also led to larger inspiratory tidal volumes in transplanted rats during a respiratory challenge (i.e., brief hypoxic exposures) compared to that in non-transplanted rats [186]. Neural precursor transplantation resulted in similarly improved outcomes. Ongoing work has revealed, however, that the preparation of cells for transplantation and their exposure to cytokines can alter the phenotypes available for transplantation. Accordingly, approaches to directing the differentiation of stem/progenitor cells toward specific neuronal phenotypes are being explored. For example, the identification of V2a spinal interneurons (glutamatergic pre-motor neurons) as a key component of the phrenic motor network [187,188] and, more recently, as a key target for improving other forms of motor recovery post-SCIs [189] has led to the development of stem-cell-derived V2a spinal interneurons for transplantation. Transplantation of V2a interneurons enriched from neural precursor cells after cervical contusion led to improved functional diaphragm recovery in rats [190].

Another cell type used to promote SCI repair is olfactory ensheathing cells. These cells have been extensively tested pre-clinically and clinically. An important consideration is that they can be derived from either the olfactory nasal mucosa or directly (more invasively) from the olfactory bulb. These two cell sources yield quite different cells that have unique properties [184]. Olfactory ecto-mesenchymal stem cells derived from the mucosa were transplanted into the cervical spinal cord to promote the repair of respiratory pathways and were reported to improve diaphragmatic and phrenic activity associated with reduced spinal inflammation 4 months after transplantation in an acute (2 days after SCI) model of C2 contusion in rats [191]. In addition, oligodendrocyte progenitor cell transplantation promoted remyelination and restoration of homeostasis in the extracellular space [192,193]. This approach also promoted functional locomotor and sensory recovery, though its effects on the respiratory system remain to be evaluated [174].

Though invasive, cell transplantation remains an appealing therapeutic. One reason for this is that, unlike activity-based therapies and neural stimulation, transplantation is not necessarily reliant on spared tissue, but can rather promote the restoration of damaged tissue. Clinical trials have demonstrated the safety of transplantation with a number of different cell types, including oligodendrocyte progenitor cells [194] and olfactory ensheathing cells [195]. Some respiratory function improvements were also observed in children with complete cervical SCIs after bone marrow nucleated cell transplantation into the spinal cord cavity with intravenous injection of these cells [196]. However, clinical information about respiratory recovery after cell transplantation remains scarce. There is hope on the horizon for cell therapies to improve breathing post-SCI with the development of new clinical trials in coming years.

### 4.2. Nerve Grafts/Nerve Transfers

The peripheral nervous system exhibits relatively greater regenerative potential than that of the adult central nervous system. This is due in part to the myelinating cells of the periphery (Schwann cells) that facilitate axonal growth, in contrast to central myelin (from oligodendrocytes), which can inhibit axonal growth [197], among other reasons. Building on early work by Tello and Ramon y Cajal, Albert Aguayo and his team showed that grafting a sciatic nerve between the brainstem and the thoracic spinal cord could create a “bridge” between the two areas. CNS axons could, indeed, grow from one side to the other one to connect neurons across the injury [197,198].

The therapeutic potential of peripheral nerve grafting has been explored in various preclinical models of SCIs, such as cervical hemisections (Figure 2). In rats, one end of the common peroneal nerve was inserted into the C2 spinal cord in a way that compromised descending respiratory pathways. The other end was left free under the skin for several months. The activity of this nerve was then recorded, and it presented a similar pattern of electrophysiological activity to that of the phrenic nerve, demonstrating that respiratory axotomized neurons could grow their axons through the graft [199]. This showed the potential of nerve grafting for acutely promoting respiratory recovery after high SCIs. The same group then showed that nerve grafts could be employed until approximately 3 weeks after a C3 hemisection to stimulate respiratory neuron axon regeneration through the grafted nerve. Treatment at later times post-injury showed less axonal growth through the grafted nerve [200].

Translation of the nerve graft technique to humans gave birth to “nerve transfer” from an intact nerve to an injured one [201]. An attempt was made in humans with cervical SCIs that occurred at the level of phMNs (C3–C5) and for which electrical diaphragmatic “pacing” could not be achieved through the phrenic nerve due to phMN death. In these individuals, end-to-end anastomose from the fourth intercostal nerve to the phrenic nerve allowed for diaphragmatic pacing once interconnections were made between the two nerves [202].

Though promising, this therapeutic approach remains highly invasive and requires removing or deriving an intact nerve from the patient to improve another modality, which could represent a drawback.

### 4.3. Harnessing the Extracellular Environment

A number of studies have aimed to reduce the inhibitory molecules expressed in the injured spinal cord by using various therapeutic agents (Figure 2) [9]. Among these inhibitory molecules, one key target is CSPG overproduction, which is mainly secreted by the glia [55,56,203]. Even if subsets of neurons can extend their axons through the fibro-glial ‘scar’ at the lesion site [34], the accumulation of CSPGs in the perineuronal net surrounding denervated neurons can limit new synaptic input, which may be needed for repair and recovery.

Selective GAG chain disruption by chondroitinase ABC administration reduced the inhibitory effects of CSPGs on active growth cones, leading to greater axonal sprouting from injured and spared neurons after SCIs [204,205,206,207]. It also modulated inflammatory processes that led to neuroprotection in the injured spinal cord, including the promotion of anti-inflammatory macrophage polarization over pro-inflammatory polarization [208]. This resulted in a pro-repair response mediated by IL-10 [209]. Injection of chondroitinase ABC specifically at the level of the perineuronal nets surrounding phMNs at a chronic time point in rats after C2 hemisection led to the sustained recovery of the previously paralyzed hemidiaphragm [124]. This was accompanied by a reduction in CSPG receptor expression and increases in the TrkB receptor expression and the density of serotonergic fibers [124]. CSPG digestion at the lesion or within the perineuronal net was also combined with other therapeutics, such as rehabilitation training and cell transplantation, in an attempt to amplify these beneficial effects [210]. For example, when combined with an IH protocol, certain molecular effects appeared to be amplified, but without further increasing the observed respiratory recovery [124]. In combination with a tibial nerve graft (one end inserted at C2 and one end inserted at C4) in C2 hemisected rats, CPSG digestion led to increased diaphragm activity, which was correlated with bulbospinal fiber regeneration [211].

A complication associated with the use of chondroitinase to enzymatically degrade CSPGs is that the enzyme is highly sensitive to changes in temperature. While ongoing work is investigating novel ways to modify the enzyme to address this issue, alternatives to the enzyme are also being explored. One such alternative is to use a soluble peptide to inhibit the signaling through CSPG receptors such as PTPσ and LAR, which also leads to diaphragm recovery associated with bulbospinal respiratory axonal regrowth [61,87,212]. Ongoing work by NervGen^®^ aims to prepare a product of this nature that can be used to treat injuries in people.

Biomaterials such as hydrogels can also be used to deliver therapeutic factors to the damaged spinal cord to promote the protection and/or plasticity of respiratory circuitry [9]. For example, ongoing work is exploring whether biomaterials can be used to provide chondroitinase stability for prolonged delivery. In addition to offering a substrate that can support tissue growth and repair, biomaterials provide an effective means of drug delivery. When used to deliver minocycline hydrochloride (an antibiotic/anti-inflammatory agent) focally to the lesion in a model of cervical contusion in rats, preservation of diaphragm activity was observed [213]. This showed that targeting inflammatory processes through interventions in the extracellular space can reduce functional loss. However, to date, these therapeutics have not been translated into clinical testing.

### 4.4. Neurotrophin Delivery

Neurotrophin expression is increased post-SCI, but not in amounts sufficient to lead to significant axonal growth, synaptic reconnectivity, or functional recovery [39]. Their potential for inducing plasticity has, however, been used to induce improvements after SCIs, such as in locomotor function [214] and respiratory function [53,215,216] (Figure 2). In particular, the BDNF/TrkB signaling pathway has been highly targeted, with a number of different delivery approaches having been tested to date.

For example, delivery of BDNF through chronic intrathecal infusion at the C4 spinal cord after C2 hemisection in rats led to diaphragm activity recovery [53]. Local BDNF hydrogel delivery in the intrathecal space after C4/C5 contusion also partially restored diaphragm function [215]. These studies showed that modulating the BDNF/TrkB signaling pathway in and/or around phMNs can sustain diaphragm activity recovery after SCIs.

In addition, viral delivery of BDNF or TrkB also showed promising results. Following C2 hemisection in rats, ipsi-lesional intraspinal (C3–C5) delivery of adeno-associated virus serotype 2 (AAV2)-BDNF vector promoted significant functional diaphragm recovery. This was correlated with the sprouting of descending bulbospinal respiratory axons and enhanced synaptic connectivity between these growing axon populations and their phMN targets [216]. A more targeted way to increase the BDNF/TrkB signaling pathway specifically in phMNs is through retrograde transduction of phMNs by using intrapleural injection of AAV vector to express TrkB receptor [217]. The resulting TrkB receptor overexpression was sufficient to increase the recovery of ipsi-lesional rhythmic phrenic activity after C2 hemisection in rats [217].

Some human studies also evaluated the effects of BDNF via systemic or intrathecal delivery. However, these BDNF administration paradigms seemed to induce several adverse effects [39]. In their review, Sieck and Mantilla suggested specifically targeting phMNs to avoid undesirable side effects [39].

## 5. Conclusions

Among the host of permanent deficits caused by SCIs, respiratory dysfunction and impaired breathing remain some of the most life-threatening. A diverse array of therapeutics have been developed and preclinically tested to reduce or reverse these deficits by targeting different aspects of the resulting pathology. However, none have been shown to be capable of promoting complete recovery, and when translated to human studies, their efficacy has remained limited. Building on a long history of therapeutic development, future preclinical studies aiming to decipher the biological processes occurring after SCIs will help in finding treatments for promoting functional respiratory recovery. Combinatorial strategies that can each target unique aspects of an injury’s pathology to enhance recovery in an additive or even synergistic manner are an important step toward achieving respiratory restoration after SCIs.

## Figures and Tables

**Figure 1 cells-12-01519-f001:**
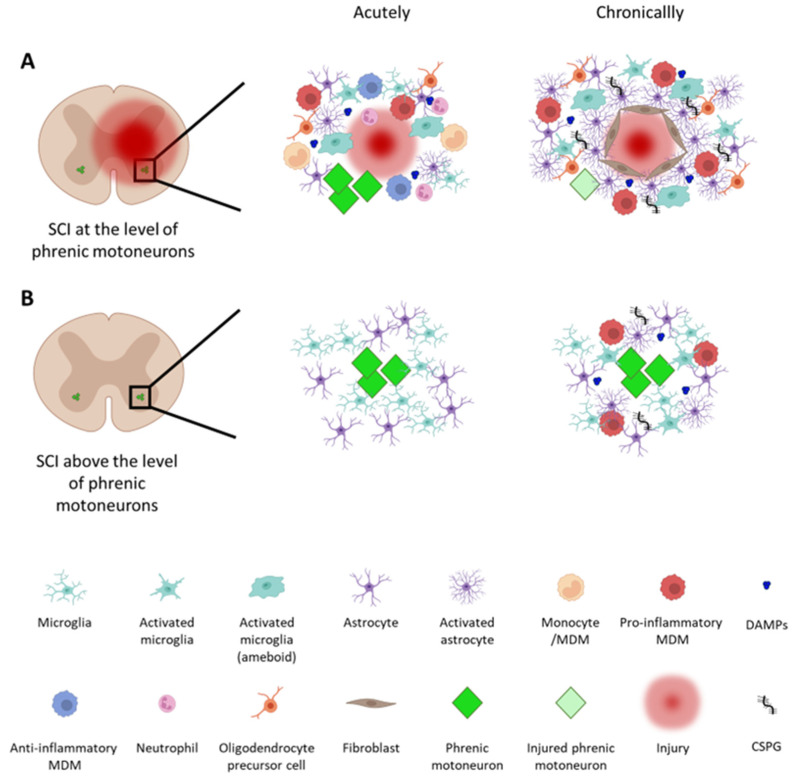
Immune response and cellular reorganization at the level of phrenic motoneurons following a traumatic cervical spinal cord injury (SCI). SCIs lead to cascades of cellular and molecular events both acutely and chronically after an injury. (**A**) When an injury occurs at the level of phrenic motoneurons (phMNs), well-known inflammatory processes occur (see the reviews in [19,33,34,35,36]). In this situation, phMNs are not only directly impacted by the injury, but are exposed to secondary inflammation, which can place them at risk of damage and/or death. (**B**) Even with an injury occurring rostral to phMNs, they are denervated and do not receive input from brainstem neurons within the ventral respiratory column (VRC). When they are only denervated, however, they typically survive and remain recruitable when spontaneous or therapeutically induced repair processes occur. MDM: monocyte-derived macrophage; DAMPs: damage-associated molecular patterns; CSPG: chondroitin sulfate proteoglycan. The figure was created with BioRender.com.

**Figure 2 cells-12-01519-f002:**
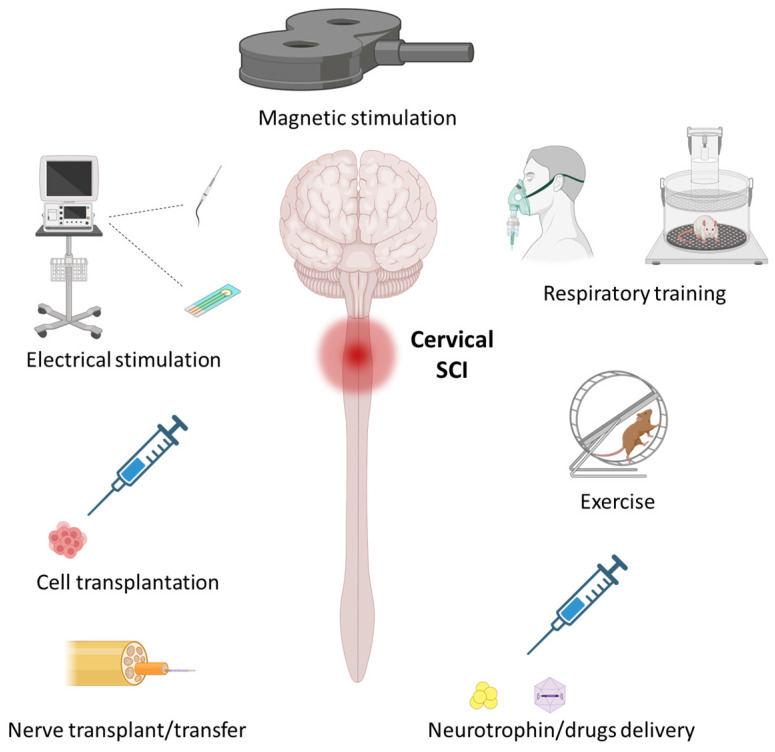
Therapeutics aimed at promoting respiratory recovery following traumatic cervical spinal cord injury (SCI). Various therapeutics have been developed to induce neuroplasticity in the phrenic motor system to promote respiratory recovery. Some of the most extensively studied are detailed in the present review and involve the following: activity-based therapeutics, such as respiratory training and exercise; stimulation-based therapeutics, such as invasive electrical stimulation and non-invasive magnetic stimulation; therapeutics for promoting axonal growth and neuroprotection, such as nerve transfer, cell transplantation, and neurotrophin/drug delivery. The figure was created with BioRender.com.

## Data Availability

Not applicable.

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
