# Peer review of "Therapeutic Strategies Targeting Respiratory Recovery after Spinal Cord Injury: From Preclinical Development to Clinical Translation"

_cells, 2023, doi:10.3390/cells12111519_

Round 1

Reviewer 1 Report

Michel-Flutot et al. have provided a thorough review of therapeutic strategies targeting the recovery of respiratory function after spinal cord injury. I have just one substantive comment: The Conclusion section was relatively brief and did little to synthesize the components of the paper. I would encourage the authors to seek a stronger impact, succintly summarizing the main points to be gleaned from this review (highlighting strengths, weaknesses, and areas requiring further work).

What was more problematic was the writing. In general, the text is well-organized and grammatical. The problem is that there were a multitude of small issues that need editing. Many simply have to do with odd word useage or repetition (same word used two or more times in close proximity). I am guessing that this occurred because the first and senior authors are not native speakers. I know that the second and third authors are excellent writers and should be able to catch problematic words/phrasing. I would strongly encourage them to carefully proof/edit the text. I am guessing too that the authors contributed distinct sections, which would explain why some portions are in need of editing and others are not. This is not a minor issue—my copy of the paper has perhaps 100 marks indicating sentences in need of repair. 

Here are some of the issues that need to be addressed (not an exhaustive list):

Many suggest ‘since’ should only be used in reference to time (substitute ‘because);

Avoid using the same word repeatedly across sentences (e.g., ‘also’ in lines 68-71 of page 2);

On line 79, ‘clues’ is correct, but an unusual usage (perhaps substitute ‘signals’);

On line 126, ‘initially physiologically extracellular’ is unclear;

On line 162, ‘As explained’ is not a common usage (consider substituting ‘As reviewed above’);

On line 173, ‘have then been’ could be ‘have been’;

On line 175, ‘high affinity several’ should be ‘several high affinity’;

Line 221, ‘pre-clinically’ could be deleted;

Line 236, ‘consisting in reducing the percentage’ is unclear;

Lines 238-239, replace ‘duration and’ with duration, ‘;

Line 245, ‘meaning it strengthened’ could be simplified to ‘strengthening’;

Line 249, ‘displayed’ is used in novel way (could substitute ‘found’);

Line 252, could delete ‘indeed’;

Line 253, ‘in rat’ should be ‘in a rat’;

Line 254, ‘displayed’ is used in novel way again (could substitute ‘showed’);

Line 260, ‘such a’ should be ‘such as a’;

Line 272, ‘two first days of protocol’ could be simplified to ‘first two days’;

Line 275, ‘to globally have some’ should be ‘to have some global’ (split infinitive); 

Line 278, could delete ‘also’;

Figure 2 caption needs to be edited;

Lines 300-308 require extensive editing; 

Lines 335-347 require extensive editing;

Line 356, ‘In similar way’ is an odd word usage (could substitute ‘Likewise’);

Lines 370-378 require extensive editing;

Line 382, ‘clinic’ could be changed to ‘clinical’;

Lines 390-391 require editing;

Lines 397-398, ‘similary vast’ could be deleted;

Line 431, ‘a growing amount of’ could be simplified to ‘increased’;

Line 442, ‘cell’ could be deleted (implied by prior sentence);

Line 474, ‘graft has therefore been used’ could be simplified to ‘grafts have been explored’;

Lines 485-488 require editing;

Line 495, ‘use it to’ can be deleted;

Line 519, ‘combined to’ should, I believe, be ‘combined with’;

Line 534, ‘Essentially’ could be deleted;

Line 566, ‘took take’ should be ‘took’.

Author Response

Comments and Suggestions for Authors

Michel-Flutot et al. have provided a thorough review of therapeutic strategies targeting the recovery of respiratory function after spinal cord injury. I have just one substantive comment: The Conclusion section was relatively brief and did little to synthesize the components of the paper. I would encourage the authors to seek a stronger impact, succintly summarizing the main points to be gleaned from this review (highlighting strengths, weaknesses, and areas requiring further work).

We thank the reviewer for this comment. We rewrite the conclusion in order to adress his comment : « Among the host of permanent deficits caused by SCI, respiratory dysfunction and impaired breathing remain some of the most life-threatening. A diverse array of therapeutics has been developed and tested preclinically to reduce or reverse these deficits by targeting different aspects of the resulting pathology. However, none have been shown capable of promoting complete recovery, and when translated to human studies, the efficacy remains limited. Building on a long history of therapeutic development, future preclinical studies aiming to decipher the biological processes occurring after SCI will help in finding treatments for promoting functional respiratory recovery. Combinatorial strategies that can each target unique aspects of injury pathology to enhance recovery in an additive or even synergistic manner is an important step toward achieving respiratory restoration after SCI. »

Comments on the Quality of English Language

What was more problematic was the writing. In general, the text is well-organized and grammatical. The problem is that there were a multitude of small issues that need editing. Many simply have to do with odd word useage or repetition (same word used two or more times in close proximity). I am guessing that this occurred because the first and senior authors are not native speakers. I know that the second and third authors are excellent writers and should be able to catch problematic words/phrasing. I would strongly encourage them to carefully proof/edit the text. I am guessing too that the authors contributed distinct sections, which would explain why some portions are in need of editing and others are not. This is not a minor issue—my copy of the paper has perhaps 100 marks indicating sentences in need of repair. 

We thank the reviewer for his comment. The manuscript has been extensively corrected by our 2 English native speakers who co-sign the manuscript.

Reviewer 2 Report

In this report the authors review the treatment strategies for respiratory recovery after cervical spinal cord injury.

The manuscript is comprehensive and well written.

This reviewer has a few suggestions.

It should be clear that section 1.1 and 1.2 regards spinal cord lesion in general, and not phrenic nerve nuclei and their descending input specifically.

In 1.1, it would be expected that loss of descending sympathetic nerve fibers would cause vasodilatation, with possible resultant edema, but not ischemia. The authors should explain this.

Figure 1 is not good and should be deleted.

In 3.1, regarding invasive stimulation, the cited percutaneous stimulation regards non-invasive stimulation.

In 3.2, non-invasive stimulation, trans-spinal direct current stimulation merits some space as a future possibility (see doi.org/10.1186/s10195-021-00623-6; doi.org/10.1186/s12984-019-0589-6).

In 4.3, replace (REF) by the necessary reference number

A few typos should be corrected, like as stimulation in 3.1 and 3.2.

Author Response

Comments and Suggestions for Authors

In this report the authors review the treatment strategies for respiratory recovery after cervical spinal cord injury.

The manuscript is comprehensive and well written.

We thank the reviewer for his positive comment.

This reviewer has a few suggestions.

It should be clear that section 1.1 and 1.2 regards spinal cord lesion in general, and not phrenic nerve nuclei and their descending input specifically.

We added a sentence to better introduce the Figure 1 and the phrenic nerve nuclei.

In 1.1, it would be expected that loss of descending sympathetic nerve fibers would cause vasodilatation, with possible resultant edema, but not ischemia. The authors should explain this.

We rewrite the sentence to make it more clearer : « Ischemic damage of intact tissues, from blood vessel disruption and any potential tissue compression (e.g., from surrounding bone), contributes to additional cell death and tissue loss. This can be exacerbated if sympathetic descending pathways are impacted and vascular function is impaired [13].» The oedema is not firstly due to a vasodilation, but to the mechanical injury. It results in vessels disruption and blood accumulation that form the oedema. Ischemia of intact tissues is the subsequent result of blood vessel irrigation loss in the surounding area.

Figure 1 is not good and should be deleted.

We would like to keep this figure, since it is an example of the main processes which occur following a cervical SCI (close or into the phrenic motoneuron area). We introduced this figure by adding a new sentence at the end of the 1.1 chapter : « For example, these processes occur around the phMN area (C3-5), which is the main topic of this review (Figure 1).  »

In 3.1, regarding invasive stimulation, the cited percutaneous stimulation regards non-invasive stimulation.

We deleted the percutaneous stimulation in this paragraph since it does not belong to it.

In 3.2, non-invasive stimulation, trans-spinal direct current stimulation merits some space as a future possibility (see doi.org/10.1186/s10195-021-00623-6; doi.org/10.1186/s12984-019-0589-6).

We added a new paragraph including the reference the reviewer suggested : « Several non-invasive stimulation procedures have been tested following cervical SCI. For example, trans-spinal direct current stimulation (TDCs) has proven to induce benefical functional improvement  [153]. Cervical trans-spinal direct current stimulation could also adress the loss of respiratory function in human [154]. »

In 4.3, replace (REF) by the necessary reference number

 Done

A few typos should be corrected, like as stimulation in 3.1 and 3.2.

Done

Round 2

Reviewer 1 Report

The authors have done an excellent job of addressing my prior concerns.